# Optimization of Vacuum-Microwave-Assisted Extraction of Natural Polyphenols and Flavonoids from Raw Solid Waste of the Orange Juice Producing Industry at Industrial Scale

**DOI:** 10.3390/molecules26010246

**Published:** 2021-01-05

**Authors:** Konstantinos Petrotos, Ioannis Giavasis, Konstantinos Gerasopoulos, Chrysanthi Mitsagga, Chryssoula Papaioannou, Paschalis Gkoutsidis

**Affiliations:** 1Department of Agrotechnology, School of Agricultural Sciences, University of Thessaly, Geopolis Campus, Periferiaki Odos Larisas Trikalon, 41500 Larisa, Greece; kosgera1@yahoo.gr (K.G.); xrpapa@teilar.gr (C.P.); gkoutsidis@teilar.gr (P.G.); 2Department of Food Science and Human Nutrition, School of Agricultural Sciences, University of Thessaly, Karditsa Campus, Terma Odou N. Temponera, 43100 Karditsa, Greece; igiavasis@teilar.gr (I.G.); cmitsanga@uth.gr (C.M.)

**Keywords:** orange pomace, vacuum-microwave-assisted extraction (VMAE), response surface optimization, historical data design, polyphenols, flavonoids, industrial scale optimization

## Abstract

Orange pomace (OP) is a solid waste produced in bulk as a byproduct of the orange juice industry and accounts for approximately 50% of the quantity of the fruits processed into juice. In numerous literature references there is information about diverse uses of orange pomace for the production of high-added-value products including production of natural antioxidant and antimicrobial extracts rich in polyphenols and flavonoids which can substitute the hazardous chemical antioxidants/antimicrobials used in agro-food and cosmetics sectors. In this work and for the first time, according to our knowledge, the eco-friendly aqueous vacuum microwave assisted extraction of orange pomace was investigated and optimized at real industrial scale in order to produce aqueous antioxidant/antimicrobial extracts. A Response Surface Optimization methodology with a multipoint historical data experimental design was employed to obtain the optimal values of the process parameters in order to achieve the maximum rates of extraction of OP total polyphenols and/or total flavonoids for economically optimum production at industrial scale. The three factors used for the optimization were: (a) microwave power (b) water to raw pomace ratio and (c) extraction time. Moreover, the effectiveness and statistical soundness of the derived cubic polynomial predictive models were verified by ANOVA.

## 1. Introduction

Orange fruit belongs to the species *Citrus sinensis* and to the family *Rutacae* and it originates from China. It is designated as sweet orange in order to be discriminated from the bitter orange which is known as *Citrus aurantium* [1] and it is cultivated in tropical and subtropical regions. In 2017, the global production of fresh orange fruits was 73 million MT and with Brazil to have share of 24% of this, and with China and India follow. According to Lohrasbi et al. [2], 33% of the global orange fruits production is industrially processed and 15 million MT of solid orange fruit waste, known as raw orange pomace, are produced, which roughly represents the 50% of the total orange fruits which are processed [3]. The orange pomace consists of peels, seeds, pulp, and segment residue [4].

In an investigation carried out by Sharma et al. [5], it was found out that orange peel could be a valuable raw material for production of a variety of high-added-value products as it contains substantial quantities of flavonoids, carotenoids, edible fibers, polyphenols, essential oils and ascorbic acid as well as sugars that could be a good substrate of various fermentations. Putnik et al. [6], reviewed these alternative valorization options for orange pomace which include production of antioxidants, edible fibres, pectin, essential oil, enzymes, bioethanol, electric acid, biogas and use of the pomace as soil improvement material or animal feed. Martinez-Hernandez et al. [7] studied in detail the techno-economic and environmental aspects of the energy valorization of orange peel waste for steam generation and concluded that this is feasible at industrial scale provided a minimum of 500 t/d orange peel processing capacity is used. According to their results, strategies such as stream energy recovery, excess ratio optimization, as well as water and value-added chemical recovery are recommended to develop synergetic relationships in the energy–water nexus of the conversion of orange peels into energy. These options can be used separately or in combination to each other targeting a desired total discharge of the solid waste of the orange juice producing industry. Regarding the average composition of orange pomace, Sulekha Gotmare & Jaya Gade [8] reported that sun-dried orange pomace consists of: moisture: 9.2 ± 0.01% *w*/*w*, crude fibers: 14.17 ± 0.36% *w*/*w*, protein: 12.43 ± 0.20% *w*/*w*, ash: 7.8 ± 0.01% *w*/*w*. carbohydrates: 52.90 ± 0.43% *w*/*w*. In addition, concerning the phenolic content of orange pomace, it contains polyphenols belonging mainly to the class of flavonoids and in particular flavones, flavonols, flavonons, isoflavons, anthocynidines and flavanols. The main flavonoids of orange fruit are hesperidin, narirutin, eriocitrin, naringin, neohesperidin, neoeriocitrin and among them hesperidin and narirutin are dominant [7,8,9]. Also, a series of short-chain bioactive phenolic acids including caffeic, chlorogenic, ferulic, sinapic and p-cumaric reported to be constituents of the phenolic profile of orange pomace [10]. The chemical structures of the main antioxidants of orange pomace are presented in Appendix A.

The extraction of high-added-value phenolics and flavonoids from orange pomace is one of the first steps of any scheme of biorefinery targeting a total discharge of this solid waste. Thus, a variety of methods, with some of them very innovative and cutting-edge, have been used for the effective extraction of them. Luengo et al. [11] successfully used pulsed electric field technology (PEF) in order to increase the yield of the extracted orange pomace phenolics. In this study, a 159% increase of the total polyphenols extraction yield and 2–3 fold increase of the main orange pomace flavonoids (hesperidin and naringin) was observed at 1–7 KV/cm and tPEF = 60 μs, 20 pulses, f = 1 Hz and a further improvement was observed as far as the voltage was increased. Ma et al. [12] & Khan et al. [13] commented on the advantages of using ultrasound-assisted extraction at 25 kHz/150 W/30 °C/15 min or even at more intensive conditions and it included to them better extraction yield in shorter time, the lower extraction temperature and the smaller eco-footprint [14]. Furthermore, Londono-Londono et al. [15] studied the aqueous ultrasound-assisted extraction of orange pomace. In addition, as it is cited by Dahmoune et al. [16] & Mhiri et al. [17] optimized microwave-assisted extraction of orange pomace at the short extraction time of 122–180 s, microwave power 200–500 W, and extraction temperature of 120–135 °C, and solid-to-liquid ratio of 1/25 to 1/30 led to polyphenols extraction yield of 12.20–15.74 mg GAE g^−1^ of dry orange pomace. Min et al. [18] & Kim et al. [19] presented the advantages of using pressurized fluid extraction for extracting phenolics from orange extracts including the ability of this method to extract the highest quantity of methoxylated flavones. The proposed extraction conditions were T = 200 °C, P = 1.4 MPa, t = 60 min. Casquete et al. [20,21] studied the extraction of orange pomace phenolics with high pressure and pointed out that the optimum operational conditions were 300 MPa 10 min or alternatively 500 MPa 3 min. They also reported that the application of high pressure led to the increased antimicrobial effectiveness of the obtained extracts against Gram-positive and Gram-negative bacteria like *Acinetobacter* and *Listeria.* Faber et al. [22] & Memduha [23] carried out experimental research on supercritical carbon dioxide extraction of orange pomace using various combinations of pressure and temperatures. The results of this studies suggest that low (25 to 35 MPa) than high (200 MPa) operating pressures are more favorable for the total antioxidant capacity of the obtained extracts while the yield of total polyphenols appears to be higher at higher pressures. The concentration of total polyphenols extracted by this technology reported to range from 18 to 27.8 mg GAE/g of dry orange pomace. In addition, the temperature that used was in the range 40–60 °C and the use of ethanol as cosolvent was found to be beneficial. A new trend for the extraction of polyphenols and flavonoids from solid pomace from the orange juice industry is the use of green deep eutectic solvents as a replacement of the hazardous and explosive organic solvents. Information about this novel trend of eco green extraction of orange pomace is given in a series of literature references [24,25,26].

According to Nayak et al. [27], microwave-assisted extraction (MAE) not only provided higher recovery of total polyphenols, but also quality phenolic compounds with rich antioxidant activity. In a comparison of MAE (Microwave-Assisted Extraction) with CSE (Conventional Solvent Extraction), UAE (Ultrasound-Assisted Extraction) and Accelerated Solvent Extraction (ASE), it was observed that the mechanism of each extraction, i.e., application of microwave or ultrasound or accelerated solvent, has its own effects on selected individual phenolic compounds. In addition, some of the major findings from this investigation support the idea of green MAE extraction, including for example: (i) substantial reduction in the processing time (122, 500, 900 and 7200 s for MAE, UAE, ASE and CSE, respectively); (ii) reduction in the extraction solvent consumption; (iii) higher extraction recovery of total polyphenol content (TPC) (at the lab-scale batch process, the yield of TPC was 356.75, 305.41, 184.72 and 301.27 kg ton^−1^ h^−1^ for MAE, UAE, ASE and CSE, respectively); (iv) in the case of MAE, microwaves are selectively absorbed by the residual water present in *C. sinensis* peels; and (v) better customer acceptance of the byproducts (peels) made through this MAE ‘‘cleaner, greener’’ extraction technology. In addition, Nguyen et al. [28] studied the effects of microwave extraction conditions on polyphenol content and antioxidant activity of pomelo extract (*Citrus maxima*). Their research had shown that continuous microwave-assisted extraction was more effective than intermittent extraction. Alcohol content, extraction time, microwave power, material-to-solvent ratio, that significantly influence the extraction process, were the parameters to be investigated. Ethanol concentration of 60%, microwave power of 300 W, microwave-assisted time of 2 min, and material-to-solvent ratio of 1:30 were significant parameters obtained in the research. Hayat et al. [29], compared CSE, UAE, and MAE of phenolic acids from mandarin peel. The results indicated that MAE gave the highest content of ferulic acid (0.239 g/100 g DW) compared with UAE (0.235 g/100 g), whereas the ferulic acid content obtained by CSE was found to be the lowest (0.205 g/100 g DW). Furthermore, to increase the kinetics and yields of extraction of polyphenols from orange peels, researchers have evaluated the potential of combining two processes such as ultrasound (US) and microwaves techniques [30].

The target of the present work is to *investigate* and *optimize, for the first time at industrial scale, and with both recovery and economic criteria*, the microwave extraction of raw orange pomace and obtain the optimum conditions in terms of microwave power, water to raw orange pomace ratio, extraction time, and with the aim to get maximum yield or productivity of polyphenols or/and flavonoids. This information can then be used in order to produce, at industrial scale and in a financially favorable way, high-added-value bioactive aqueous orange pomace extracts and thus to exploit a mass-produced solid agro-food waste by the orange juice production industry. The optimized aqueous orange pomace extracts can be used either as natural antioxidants or/and as natural antimicrobials (extracts with maximum polyphenols and/or flavonoids content) targeting food, nutraceutical, cosmetic and agro-protection applications and leading to the improvement of the carbon footprint of the orange juice production industry.

## 2. Results

### 2.1. Predictive Modeling and Optimization of the Extracted Amount of Orange Pomace (OP) Total Polyphenols

The results of the total polyphenol content of the orange extracts are presented in Table 1. In particular, 80 vacuum microwave extraction experiments were carried out and three samples of the extract were collected in each respective run. The total polyphenol contents of each one of the obtained three samples per run were determined and the average values of them are listed in Table 1. By using the Design Expert software Version 12.0.0 the data of Table 1 were analyzed by RSM (Surface Response Methodology) and the mathematical modeling yielded the following model equation for the prediction of the extracted OP total polyphenols (TPE) (Equation (1)):(1)1SqrtTOTAL POLYPHENOL in mg GAE per 2 Kg raw OP  =+0.015117−2.15515×10−6×MICROWAVE POWER  −0.000104×WATER TO POMACE RATIO−0.000027×EXTRACTION TIME   +3.52786×10−8×MICROWAVE POWER ×WATER TO POMACE RATIO  +6.48576×10−9×MICROWAVE POWER ×EXTRACTION TIME  −2.88249×10−7×WATER TO POMACE RATIO × EXTRACTION TIME  +3.69373×10−10×MICROWAVE POWER2−1.28659×10−6×WATER TO POMACE RATIO2  +1.01714×10−7×EXTRACTION TIME2  −1.36113×10−11×MICROWAVE POWER×2WATER TO POMACE RATIO  −1.00364×10−12×MICROWAVE POWER×2EXTRACTION TIME  +1.54571×10−9×MICROWAVE POWER × WATER TO POMACE RATIO2

Furthermore, the ANOVA statistics of the RSM model, derived to correlate the extracted OP total polyphenols with the experimental factors, are presented in Table 2. Following ANOVA results it is concluded that the model is significant while its lack of fit is not significant. This implies that the developed model is good tool for prediction of the extracted amount of OP total polyphenols vs. the three experimental factors: (a) Microwave power (W) (b) Water to orange pomace ratio and (c) Extraction time (min) within the experimental design space. In addition, the calculated R^2^ value = 0.8417 is satisfactory and simultaneously there is a high degree of proximity between the calculated values of Adjusted R^2^ = 0.8133 and Predicted R^2^ = 0.7705 as the difference between them is, by far, less than the limit of 0.2 set by the method. Therefore, the derived RSM model can be used for effective prediction of the extracted amount of OP polyphenols within the design space as well as for optimization.

The robustness of the derived model is further supported by the data presented in Figure 1 which illustrates the correlation among the predicted and actual values of the total amount of the extracted OP polyphenols and all the points are lying close to the central linear plot (45° line). Furthermore, in the Figure 2, Figure 3 and Figure 4 the effect of the paired interactions of the extraction factors A = microwave power, B = water/solid ratio, (W), C = extraction time (min) on the OP total polyphenols extraction yield is presented. Finally, by using the derived RSM cubic model and the facility of Expert Design 12.0.0 statistical software, the maximum value of the extracted OP total polyphenols was obtained at the following condition:Microwave power: 5999.997 W,Water to orange pomace ratio = 26.09Extraction Time: 120.00 min

Whereas, the corresponding maximum amount of extracted OP total polyphenols, at the above-mentioned conditions, was found to be 13,559.802 mg per 2 Kg of extracted raw orange pomace.

**Table 1 molecules-26-00246-t001:** Amount of total polyphenols and total flavonoids in raw orange pomace (OP) extracts and calculated productivity indices.

A/A	Microwave Power	*** Water-to-Solid Ratio	Extraction Time (min)	* Amount of Total Polyphenols in the Extract Expressed as Gallic Acid Equivalents (mg GA)	** Amount of Total Flavonoids in the OP Extract Expressed as Quercetin Equivalent (mg QE)	Rate of Extraction of OP Total Polyphenols (mg GAE Kg^−1^ min^−1^)	Rate of Extraction of Orange Pomace Total Flavonoids (mg QE Kg^−1^ min^−1^)
**1**	4000.00	20.00	15.00	8480 ± 121	630.3 ± 58	141.333 ± 2.02	10.505 ± 0.97
**2**	4000.00	20.00	30.00	9280 ± 132	877.58 ± 101	103.111 ± 1.47	9.75089 ± 1.12
**3**	4000.00	20.00	45.00	9760 ± 98	1229.09 ± 124	81.3333 ± 0.82	10.2424 ± 1.03
**4**	4000.00	20.00	60.00	10,240 ± 82	1517.58 ± 134	68.2667 ± 0.55	10.1172 ± 0.89
**5**	4000.00	20.00	75.00	10,480 ± 93	1949.09 ± 121	58.2222 ± 0.52	10.8283 ± 0.67
**6**	4000.00	20.00	90.00	10,800 ± 112	710.3 ± 76	51.4286 ± 0.53	3.38238 ± 0.36
**7**	4000.00	20.00	120.00	11,040 ± 65	576.97 ± 91	40.8889 ± 0.24	2.13693 ± 0.34
**8**	2000.00	30.00	15.00	8160 ± 84	690.91 ± 61	136 ± 1.40	11.5152 ± 1.02
**9**	2000.00	30.00	30.00	8640 ± 105	899.39 ± 72	96 ± 1.17	9.99322 ± 0.80
**10**	2000.00	30.00	45.00	9520 ± 132	1231.52 ± 123	79.3333 ± 1.10	10.2627 ± 1.03
**11**	2000.00	30.00	60.00	9920 ± 64	1481.21 ± 115	66.1333 ± 0.43	9.87473 ± 0.77
**12**	2000.00	30.00	75.00	9600 ± 76	1922.42 ± 131	53.3333 ± 0.42	10.6801 ± 0.73
**13**	2000.00	30.00	90.00	10,720 ± 123	768.48 ± 76	51.0476 ± 0.59	3.65943 ± 0.36
**14**	2000.00	30.00	120.00	10,800 ± 102	620.61 ± 73	40 ± 0.38	2.29856 ± 0.27
**15**	6000.00	30.00	15.00	8320 ± 78	218.18 ± 43	138.667 ± 1.30	3.63633 ± 0.72
**16**	6000.00	30.00	30.00	10,080 ± 103	300.61 ± 34	112 ± 1.14	3.34011 ± 0.38
**17**	6000.00	30.00	45.00	10,400 ± 86	409.7 ± 51	86.6667 ± 0.72	3.41417 ± 0.43
**18**	6000.00	30.00	60.00	10,400 ± 58	705.45 ± 112	69.3333 ± 0.39	4.703 ± 0.75
**19**	6000.00	30.00	75.00	10,080 ± 89	375.76 ± 43	56 ± 0.49	2.08756 ± 0.24
**20**	6000.00	30.00	90.00	12,080 ± 104	460.61 ± 52	57.5238 ± 0.50	2.19338 ± 0.25
**21**	6000.00	30.00	120.00	12,800 ± 134	460.61 ± 47	47.4074 ± 0.50	1.70596 ± 0.17
**22**	4000.00	30.00	15.00	8640 ± 87	630.3 ± 71	144 ± 1.45	10.505 ± 1.18
**23**	4000.00	30.00	30.00	9040 ± 97	295.76 ± 22	100.444 ± 1.08	3.28622 ± 0.24
**24**	4000.00	30.00	45.00	9760 ± 101	317.58 ± 41	81.3333 ± 0.84	2.6465 ± 0.34
**25**	4000.00	30.00	60.00	9760 ± 69	378.18 ± 52	65.0667 ± 0.46	2.5212 ± 0.35
**26**	4000.00	30.00	75.00	10,400 ± 114	375.76 ± 43	57.7778 ± 0.63	2.08756 ± 0.24
**27**	4000.00	30.00	90.00	11,040 ± 87	392.73 ± 52	52.5714 ± 0.41	1.87014 ± 0.25
**28**	4000.00	30.00	120.00	10,800 ± 93	450.91 ± 23	40 ± 0.34	1.67004 ± 0.09
**29**	2000.00	10.00	15.00	7680 ± 111	173.33 ± 25	128 ± 1.85	2.88883 ± 0.42
**30**	2000.00	10.00	30.00	8040 ± 132	214.55 ± 31	89.3333 ± 1.47	2.38389 ± 0.34
**31**	2000.00	10.00	45.00	8320 ± 86	200 ± 23	69.3333 ± 0.72	1.66667 ± 0.19
**32**	2000.00	10.00	60.00	8440 ± 75	256.97 ± 33	56.2667 ± 0.50	1.71313 ± 0.22
**33**	2000.00	10.00	90.00	8960 ± 83	260.61 ± 21	42.6667 ± 0.40	1.241 ± 0.10
**34**	2000.00	10.00	120.00	8680 ± 64	246.06 ± 26	32.1481 ± 0.24	0.911333 ± 0.10
**35**	4000.00	10.00	15.00	9600 ± 73	272.73 ± 41	160 ± 1.22	4.5455 ± 0.68
**36**	4000.00	10.00	30.00	8880 ± 101	305.45 ± 15	98.6667 ± 1.12	3.39389 ± 0.17
**37**	4000.00	10.00	45.00	9000 ± 85	443.64 ± 25	75 ± 0.71	3.697 ± 0.21
**38**	4000.00	10.00	60.00	9960 ± 91	461.82 ± 27	66.4 ± 0.61	3.0788 ± 0.18
**39**	4000.00	10.00	75.00	10,320 ± 121	469.09 ± 36	57.3333 ± 0.67	2.60606 ± 0.20
**40**	4000.00	10.00	90.00	11,160 ± 112	469.09 ± 41	53.1429 ± 0.53	2.23376 ± 0.20
**41**	6000.00	20.00	15.00	8760 ± 103	261.82 ± 27	146 ± 1.72	4.36367 ± 0.45
**42**	6000.00	20.00	30.00	9960 ± 85	247.27 ± 32	110.667 ± 0.94	2.74744 ± 0.36
**43**	6000.00	20.00	45.00	10,800 ± 126	363.64 ± 21	90 ± 1.05	3.03033 ± 0.18
**44**	6000.00	20.00	75.00	12,360 ± 132	341.82 ± 26	68.6667 ± 0.73	1.899 ± 0.14
**45**	6000.00	20.00	90.00	13,320 ± 98	374.55 ± 32	63.4286 ± 0.47	1.78357 ± 0.15
**46**	6000.00	20.00	120.00	13,800 ± 95	334.55 ± 41	51.1111 ± 0.35	1.23907 ± 0.15
**47**	6000.00	10.00	15.00	6560 ± 76	133.33 ± 21	109.333 ± 1.27	2.22217 ± 0.35
**48**	6000.00	10.00	30.00	7640 ± 81	178.18 ± 11	84.8889 ± 0.90	1.97978 ± 0.12
**49**	6000.00	10.00	45.00	7480 ± 73	189.09 ± 16	62.3333 ± 0.61	1.57575 ± 0.13
**50**	6000.00	10.00	60.00	8080 ± 97	192.73 ± 21	53.8667 ± 0.65	1.28487 ± 0.14
**51**	6000.00	10.00	75.00	7640 ± 71	220.61 ± 31	42.4444 ± 0.39	1.22561 ± 0.17
**52**	6000.00	10.00	90.00	8240 ± 104	318.79 ± 11	39.2381 ± 0.50	1.51805 ± 0.05
**53**	6000.00	10.00	120.00	8600 ± 73	233.94 ± 13	31.8519 ± 0.27	0.866444 ± 0.05
**54**	4000.00	20.00	15.00	8560 ± 86	613.33 ± 16	142.667 ± 1.43	10.2222 ± 0.27
**55**	4000.00	20.00	30.00	9280 ± 91	836.36 ± 32	103.111 ± 1.01	9.29289 ± 0.36
**56**	4000.00	20.00	45.00	9120 ± 94	1236.36 ± 37	76 ± 0.78	10.303 ± 0.31
**57**	4000.00	20.00	60.00	10,240 ± 128	1541.82 ± 82	68.2667 ± 0.85	10.2788 ± 0.55
**58**	4000.00	20.00	75.00	10,080 ± 121	1927.27 ± 56	56 ± 0.67	10.7071 ± 0.31
**59**	4000.00	20.00	90.00	11,040 ± 89	671.52 ± 31	52.5714 ± 0.42	3.19771 ± 0.15
**60**	4000.00	20.00	120.00	10,560 ± 115	545.45 ± 21	39.1111 ± 0.43	2.02019 ± 0.08
**61**	2000.00	20.00	15.00	9200 ± 123	749.09 ± 72	153.333 ± 2.05	12.4848 ± 1.20
**62**	2000.00	20.00	30.00	9120 ± 111	1129.7 ± 103	101.333 ± 1.23	12.5522 ± 1.14
**63**	2000.00	20.00	45.00	9440 ± 87	1343.03 ± 97	78.6667 ± 0.73	11.1919 ± 0.81
**64**	2000.00	20.00	60.00	9360 ± 91	1575.76 ± 84	62.4 ± 0.61	10.5051 ± 0.56
**65**	2000.00	20.00	75.00	11,200 ± 115	1934.55 ± 115	62.2222 ± 0.64	10.7475 ± 0.64
**66**	2000.00	20.00	90.00	10,560 ± 111	683.64 ± 52	50.2857 ± 0.53	3.25543 ± 0.25
**67**	2000.00	20.00	120.00	10,880 ± 87	656.97 ± 23	40.2963 ± 0.32	2.43322 ± 0.09
**68**	4000.00	20.00	15.00	9280 ± 121	673.94 ± 28	154.667 ± 2.02	11.2323 ± 0.47
**69**	4000.00	20.00	30.00	9360 ± 114	1095.76 ± 104	104 ± 1.27	12.1751 ± 1.16
**70**	4000.00	20.00	45.00	9680 ± 74	1318.79 ± 121	80.6667 ± 0.62	10.9899 ± 1.01
**71**	4000.00	20.00	60.00	10,160 ± 102	1604.85 ± 140	67.7333 ± 0.68	10.699 ± 0.93
**72**	4000.00	20.00	75.00	10,720 ± 116	1854.55 ± 117	59.5556 ± 0.64	10.3031 ± 0.65
**73**	4000.00	20.00	90.00	10,240 ± 121	722.42 ± 41	48.7619 ± 0.58	3.4401 ± 0.20
**74**	4000.00	20.00	120.00	10,800 ± 117	686.06 ± 52	40 ± 0.43	2.54096 ± 0.19
**75**	4000.00	20.00	30.00	10,680 ± 89	334.55 ± 21	118.667 ± 0.99	3.71722 ± 0.23
**76**	4000.00	20.00	45.00	11,040 ± 113	378.18 ± 19	92 ± 0.94	3.1515 ± 0.16
**77**	4000.00	20.00	60.00	11,040 ± 121	560 ± 18	73.6 ± 0.81	3.73333 ± 0.12
**78**	4000.00	20.00	75.00	10,440 ± 109	658.18 ± 41	58 ± 0.61	3.65656 ± 0.23
**79**	4000.00	20.00	90.00	11,760 ± 118	530.91 ± 38	56 ± 0.56	2.52814 ± 0.18
**80**	4000.00	20.00	120.00	11,280 ± 76	450.91 ± 33	41.7778 ± 0.28	1.67004 ± 0.12

* The Figures of this column are calculated by multiplying the total polyphenols concentration of the extract by its volume = CP (mg/L) × Water-to-solid ratio × 2 Kg. ** The figures of this column are calculated by multiplying the total flavonoids concentration of the extract by its volume = CF (mg/L) × water-to-solid ratio × 2 Kg. *** Water/solid ratio is expressed in L Kg^−1^ and the mass of the solids were equal to 2 Kg for all experiments. **** The values in columns 4,5,6,7 are the average ± SD (standard deviation).

**Table 2 molecules-26-00246-t002:** Analysis of Variance (ANOVA) of the derived model for prediction of the amount of the total polyphenols in OP extracts.

	Sum of Squares	df	Mean Square	F-Value	*p*-Value	
**Model**	0.0000	12	2.694 × 10^−6^	29.68	**<0.0001**	**s** **ignificant**
A—MICROWAVE POWER	1.232 × 10^−6^	1	1.232 × 10^−6^	13.57	**0.0005**	
B—WATER-TO-POMACE RATIO	1.453 × 10^−8^	1	1.453 × 10^−8^	0.1601	0.6903	
C—EXTRACTION TIME	3.906 × 10^−6^	1	3.906 × 10^−6^	43.04	**<0.0001**	
AB	1.492 × 10^−6^	1	1.492 × 10^−6^	16.44	**0.0001**	
AC	4.464 × 10^−7^	1	4.464 × 10^−7^	4.92	**0.0300**	
BC	3.538 × 10^−7^	1	3.538 × 10^−7^	3.90	0.0525	
A²	2.789 × 10^−7^	1	2.789 × 10^−7^	3.07	0.0842	
B²	4.149 × 10^−6^	1	4.149 × 10^−6^	45.71	**<0.0001**	
C²	9.562 × 10^−7^	1	9.562 × 10^−7^	10.54	**0.0018**	
A²B	2.560 × 10^−6^	1	2.560 × 10^−6^	28.21	**<0.0001**	
A²C	3.512 × 10^−7^	1	3.512 × 10^−7^	3.87	0.0533	
AB²	8.331 × 10^−7^	1	8.331 × 10^−7^	9.18	**0.0035**	
**Residual**	6.081 × 10^−6^	67	9.076 × 10^−8^			
Lack of Fit	4.673 × 10^−6^	47	9.942 × 10^−8^	1.41	0.2024	**not significant**
Pure Error	1.408 × 10^−6^	20	7.042 × 10^−8^			
**Cor Total**	0.0000	79				

Type of polynomial Model: Reduced Cubic model**Response:** AMOUNT OF EXTRACTED OP TOTAL POLYPHENOLS (mg/2 kg raw OP)**Transform:** Inverse SqrtConstant: 0Fit Statistics


**Std. Dev.**
0.0003
**R²**
0.8417
**Mean**
0.0101
**Adjusted R²**
0.8133
**C.V. %**
2.97
**Predicted R²**
0.7705


**Adeq Precision**
27.7094

### 2.2. Predictive Modeling and Optimization of the Extracted Amount of OP Total Flavonoids

The results of the extracted amount of OP total flavonoids are summarized in Table 1. In particular, 80 vacuum-microwave-assisted extraction experiments were carried out and three samples of the extract were collected per each respective run. The total flavonoid content of each one of the three collected extracts per run were determined and the average values of them are listed in Table 1. By using the Design Expert software Version 12.0.0 the OP total flavonoids data were analyzed by RSM (Surface Response Methodology) and the mathematical modeling yielded to the Equation (2), which correlates the amount of the extracted OP total flavonoids (TFE), expressed as quercetin equivalents (QE), with the three respective experimental factors: (a) Microwave power (W) (b) Water-to-OP ratio and (c) Extraction time (min). Furthermore, by using the guidance of the Cox and Box graph, the inverse squared option was selected as the most appropriate transformation of the OP total flavonoids response for effective data fitting and consequently by regression the cubic polynomial predictive model equation was derived (Equation (2)):(2)$$\begin{array}{c} \frac{1.0}{Sqrt\left(TOTAL\;FLAVONOIDS\;in\;mg\;QE\;per\;2\;Kg\;of\;raw\;OP\right)}\;\\ \;=+0.30080-7.64856\;X10^{-5}\times \;MICROWAVE\;POWER\;\\ \;-0.019707\;\;\;\;\;\;\;\times WATER\;TO\;POMACE\;RATIO-8.02145\;X\;10^{-4}\times \;EXTRACTION\;TIME\\ \;+4.25803\;X\;10^{-6}\times \;MICROWAVE\;POWER\;\times WATER\;TO\;POMACE\;RATIO\\ \;-3.60410\;X\;10^{-8}\times \;MICROWAVE\;POWER\;\times \;EXTRACTION\;TIME\\ \;+4.92113\;X\;10^{-5}\times \;WATER\;TO\;POMACE\;RATIO\times \;EXTRACTION\;TIME\;\\ \;+8.53348\;X\;10^{-9}\times MICROWAVE\;POWER^{2}\\ \;+3.27486\;X\;10^{-4}\times WATER\;TO\;POMACE\;RATIO^{2}+3.03073\;X\;10^{-6}\times \;EXTRACTION\;TIME^{2}\\ \;-3.48160\;X\;10^{-10}\times \;MICROWAVE\;POWER^{2}\times WATER\;TO\;POMACE\;RATIO\\ \;-3.22215\;X\;10^{-8}\times \;MICROWAVE\;POWER\;\times \;WATER\;TO\;POMACE\;RATIO^{2}\\ \;-1.09277\;X\;10^{-6}\times \;WATER\;TO\;POMACE\;RATIO^{2}\times \;EXTRACTION\;TIME \end{array}$$

Moreover, in Table 3, the model’s ANOVA statistics are presented. According to ANOVA, the model is significant while its lack of fit is not significant which implies that it is a good tool for prediction of the amount of total OP flavonoids in the extracts vs. the three experimental factors within the selected design space. The R^2^ value (0.8552) is high, which reveals a satisfactory correlation of the extracted amount of OP total flavonoids with the three experimental factors. In addition, the value of the Adjusted R^2^ (0.8293) is very close to the Predicted R^2^ (0.8105) value, which means that there is satisfactory prediction by the developed model equation. The effectiveness of derived model is additionally supported by the data presented in Figure 5. In this Figure, the predicted vs. actual values of the amount of the extracted OP total flavonoids are correlated and all the individual points come very close to the central linear plot with an angle of 45°. In addition, in the Figure 6, Figure 7 and Figure 8, the plots illustrate the effect of the paired interactions of the three factors (A = Microwave power (W), B = Water/solid ratio, C = Extraction time (min)) on the orange pomace total flavonoids extraction yield. Finally, by using the derived cubic polynomial RSM model and the facility of Expert Design 12.0.0 statistical package, the maximum value of the extracted amount OP total flavonoids was found to be reached at the following extraction condition:Microwave power: 2000 WattWater to solid orange pomace ratio: 24.12Extraction time: 53.45 min

And, the maximum amount of extracted OP total flavonoids, at the above mentioned optimum conditions, was found to be 1909.27 mg QE/2 kg of extracted raw OP.

**Table 3 molecules-26-00246-t003:** Analysis of variance (ANOVA) of the derived model for prediction of the amount of the total flavonoids in OP extracts.

ANOVA for Response Surface						
Analysis of Variance Table [Partial Sum of Squares-Type III]						
Source	Sum of Squares	df	Mean Square	F Value	*p*-Value	
Model	0.016	12	1.30	32.98	**<0.0001**	**Significant**
A—MICROWAVE POWER	1.86	1	1.86	47.37	**<0.0001**	
B—WATER TO POMAVE RATIO	4.00 × 10^−3^	1	4.00 × 10^−3^	0.10	0.7506	
C—EXTRACTION TIME	1.19 × 10^−3^	1	1.19 × 10^−3^	0.030	0.8624	
AB	3.63 × 10^−1^	1	3.63 × 10^−1^	9.24	**0.0034**	
AC	2.43 × 10^−1^	1	2.43 × 10^−1^	6.19	**0.0153**	
BC	1.29 × 10^−1^	1	1.29 × 10^−1^	3.27	0.0749	
A^2^	6.82 × 10^−1^	1	6.82 × 10^−1^	17.35	**<0.0001**	
B^2^	2.79	1	2.79	70.92	**<0.0001**	
C^2^	8.46 × 10^−1^	1	8.46	21.52	**<0.0001**	
A^2^B	1.68	1	1.68	42.65	**<0.0001**	
AB^2^	3.62 × 10^−1^	1	3.62 × 10^−1^	9.21	**0.0034**	
B^2^C	2.60 × 10^−1^	1	2.60 × 10^−1^	6.60	**0.0124**	
Residual	2.63	67	3.93 × 10^−2^			
Lack of Fit	1.38	47	2.94 × 10^−2^	0.47	0.9826	**Not Significant**
Pure Error	1.25	20	6.25 × 10^−2^			
Cor Total	0.018	79				

Type of polynomial model: reduced cubic modelResponse: AMOUNT OF EXTRACTED OP TOTAL FLAVONOIDS (mg QE per 2 kg raw OP)**Transform:** Inverse SqrtConstant: 0Fit Statistics


**Std. Dev.**
6.27R-Squared
**0.8552**

**Mean**
0.046Adj R-Squared
**0.8293**

**C.V. %**
13.72Pred R-Squared
**0.8105**
PRESS3.45Adeq Precision
**23.712**


### 2.3. Simultaneous Optimization of Total Polyphenols Content and Total Flavonoids Content of OP Extracts

In addition, the simultaneous maximization of the amount of extracted OP total polyphenol and total flavonoids was carried out. The methodology to obtain the optimum amount of the extracted OP total polyphenols and flavonoids simultaneously, was based on the function of Design Expert software that can calculate the values of the extraction parameters which lead to a common convergence of the extracted amounts of OP total polyphenols and total flavonoids to values as close as possible to their independent optima.

By this methodology, the following set of optimum extraction conditions that simultaneously maximize the extracted amounts of OP total polyphenols and OP total flavonoids was obtained:Microwave Power: 2000.00 WWater to solid orange pomace ratio: 24.95Extraction Time: 63.96 min

At the above-mentioned optimum extraction conditions for simultaneous optimization of the amount of OP total polyphenols and total flavonoids in the orange pomace extracts obtained by VMAE, the corresponding optimum values found to be 9977.48 mg GAE for total polyphenol, which represents the 73.58% of the independent polyphenols optimum, and 1847.42 mg of QE for total flavonoids, which represents the 96.76% of the independent flavonoids optimum respectively.

The predicted optimum extraction conditions and the corresponding optimum values of responses (extracted amounts of OP polyphenols and flavonoids) independently or in combination of two are summarized in Table 4.

### 2.4. The Economic Optimization of the OP VMA Extraction at Industrial Scale and the Determination of the Corresponding Optimum Values of Extraction Conditions to Obtain Maximum Rates of Extraction (Productivities) for OP Total Polyphenols and OP Total Flavonoids Respectively

#### 2.4.1. Maximization of the Rate of the Extraction (Productivity) of Raw OP Total Polyphenols

By using the derived Equation (7) in paragraph 4.6, the extraction rate of orange pomace total polyphenols was calculated for each one of the 80 experimental points and recorded in Table 1 vs. the three experimental factors (a) Microwave power (W), (b) Water to orange pomace ratio and (c) Extraction time (min). Consequently, the recorded data were introduced in Design Expert software and analyzed by adopting the multipoint historical data design and Surface Response Methodology (RSM) in order to derive the predictive polynomial model to fit them. By using the Box-Cox plot for power transforms the natural log transformation of the dependent variable (Rate of total polyphenols extraction) was selected as the best for achieving the most effective data fitting. Moreover, by using the derived model, the optimum process parameters, which maximize the extraction rate of OP total polyphenols and therefore the productivity, were determined at industrial scale. Finally, the Analysis of Variance (Table 5.) applied to the data yielded a high correlation factor R^2^ = 0.9823 and also to a high degree of proximity among the adjusted R^2^ = 0.9788 and predicted R^2^ = 0.9737 (difference by far less than the proposed maximum acceptable limit of 0.2) which implies that the derived model can be successfully used for prediction within the selected design space as well as for effective industrial scale economic optimization.

**Table 5 molecules-26-00246-t005:** Analysis of variance (ANOVA) for the reduced cubic RSM model of OP total polyphenols productivity.

Source	Sum of Squares	df	Mean Square	F-Value	*p*-Value	
Model	13.00	13	0.9997	281.68	**<0.0001**	**S** **ignificant**
A-MICROWAVE POWER	0.0551	1	0.0551	15.53	**0.0002**	
B-WATER TO OP RATIO	0.0007	1	0.0007	0.2030	0.6538	
C-EXTRACTION TIME	1.12	1	1.12	314.89	**<0.0001**	
AB	0.0540	1	0.0540	15.22	**0.0002**	
AC	0.0200	1	0.0200	5.64	**0.0205**	
BC	0.0183	1	0.0183	5.15	**0.0265**	
A²	0.0070	1	0.0070	1.96	0.1662	
B²	0.1607	1	0.1607	45.28	**<0.0001**	
C²	0.6004	1	0.6004	169.18	**<0.0001**	
A²B	0.0921	1	0.0921	25.95	**<0.0001**	
A²C	0.0136	1	0.0136	3.84	0.0544	
AB²	0.0336	1	0.0336	9.47	**0.0030**	
C³	0.0720	1	0.0720	20.29	**<0.0001**	
Residual	0.2342	66	0.0035			
Lack of Fit	0.1770	46	0.0038	1.34	0.2394	**Not Significant**
Pure Error	0.0573	20	0.0029			
Cor Total	13.23	79				

Response: TOTAL OP POLYPHENOLS PRODUCTIVITY (mg GAE kg^−1^ min^−1^)Transform: Natural LogConstant: 0

In addition, the derived RSM polynomial model equation was found to be of reduced cubic form and it is analytically given here (Equation (3))


(3)
$$\begin{array}{cc} \text{Ln}(\text{POLYPHENOLS PRODUCTIVITY in mg GAE}{\mathrm{kg}}^{-1}\text{raw OP}{\text{min}}^{-1})=\\ \text{+4.68567} & \\ \text{+0.000399} & \text{× MICROWAVE POWER}\\ \text{+0.017730} & \text{× WATER TO POMACE RATIO}\\ \text{−0.033546} & \text{× EXTRACTION TIME}\\ \text{−(6.00198 ×}10^{-6}) & \text{× MICROWAVE POWER × WATER TO POMACE RATIO}\\ \text{−(1.25585 ×}10^{-6}) & \text{× MICROWAVE POWER × EXTRACTION TIME}\\ \text{+0.000066} & \text{× WATER TO POMACE RATIO × EXTRACTION TIME}\\ \text{−(6.95551 ×}10^{-8}) & \text{× MICROWAVE }{\text{POWER}}^{2}\\ \text{+0.000278} & \text{× WATER TO POMACE }{\text{RATIO}}^{2}\\ \text{+0.000285} & \text{× EXTRACTION}{\text{TIME}}^{2}\\ \text{+(2.58285 ×}10^{-9}) & \text{× MICROWAVE}{\text{POWER}}^{2}\text{× WATER TO POMACE RATIO}\\ \text{+(1.97819 ×}10^{-10}) & \text{× MICROWAVE}{\text{POWER}}^{2}\text{× EXTRACTION TIME}\\ \text{−(3.10450 ×}10^{-7}) & \text{× MICROWAVE POWER × WATER TO POMACE}{\text{RATIO}}^{2}\\ \text{−(1.02940 ×}10^{-6}) & \text{× EXTRACTION}{\text{TIME}}^{3} \end{array}$$



Furthermore, the optimum extraction conditions and the corresponding maximum value of the raw OP total polyphenols extraction rate (productivity) were found to be the following:Microwave power: 4484.777 WWater to solid orange pomace ratio: 19.619Extraction Time: 15.00 minThe optimum rate of OP total polyphenols extraction (maximum productivity) 154.074 mg GAE Kg^−1^ min^−1^.

#### 2.4.2. Maximization of the Rate of the Extraction (Productivity) of Raw OP Total Flavonoids

By using the Equation (8), which was derived in paragraph 4.6., the rate of extraction of orange pomace total flavonoids was calculated for each one of the 80 experimental points and the calculated values are listed in Table 1 vs. the three experimental factors (a) Microwave power (W), (b) Water to orange pomace ratio and (c) Extraction time (min). Consequently, the data (as average values) were introduced in Design expert software and analyzed by adopting the multipoint historical data design and Response Surface Methodology (RSM) in order to derive the predictive cubic polynomial model to fit them. Moreover, in a subsequent step, by using the derived model, the optimum process parameters which maximize the rate of extraction of OP total flavonoids and therefore correspond to the most economical operation at industrial scale, were determined (point of maximum total OP flavonoids productivity). By consulting the Box-Cox plot for power transformations, the inverse square transformation of the dependent variable (rate of OP total flavonoids extraction) was found to be the appropriate for the most effective data fitting. In addition, Analysis of Variance (Table 6) which applied to the model presented a high correlation factor R^2^ = 0.8784 and also a high degree of proximity of the Adjusted R^2^ = 0.8627 and Predicted R^2^ = 0.8500, which implies that the derived model can be successfully used for prediction purposes within the selected design space as well as for effective optimization at industrial scale.

**Table 6 molecules-26-00246-t006:** ANOVA for reduced cubic RSM model of OP total flavonoids productivity.

ANOVA for Response Surface Reduced Cubic Model						
Analysis of Variance Table [Partial Sum of Squares-Type III]						
Source	Sum of Squares	df	Mean Square	F Value	*p*-Value	
Model	2.86	9	0.32	56.17	**<0.0001**	**S** **ignificant**
A—MICROWAVE POWER	0.26	1	0.26	45.33	**<0.0001**	
B—WATER TO POMACE RATIO	4.61 × 10^−1^	1	4.61 × 10^−1^	0.081	0.7762	
C—EXTRACTION TIME	0.99	1	0.99	175.59	**<0.0001**	
AB	0.047	1	0.047	8.22	**0.0055**	
A^2^	0.080	1	0.080	14.07	**0.0004**	
B^2^	0.41	1	0.41	72.45	**<0.0001**	
C^2^	0.050	1	0.050	8.75	**0.0042**	
A^2^B	0.24	1	0.24	41.80	**<0.0001**	
AB^2^	0.052	1	0.052	9.13	**0.0035**	
Residual	0.40	70	5.66			
Lack of Fit	0.23	50	4.62	0.56	0.9505	**Not Significant**
Pure Error	0.17	20	8.27			
Cor Total	3.26	79				

Response: OP TOTAL FLAVONOIDS PRODUCTIVITY (mg QE kg^−1^ min^−1^)Transform: Inverse Sqrt

Std. Dev.0.075R-Squared0.8784Mean0.55Adj R-Squared0.8627C.V. %13.77Pred R-Squared0.8500PRESS0.49Adeq Precision31.686

In addition, the derived RSM polynomial model equation was of reduced cubic form and it is analytically given here by Equation (4):(4)$$\frac{1.0}{Sqrt\;of\;FLAVONOIDS\;PRODUCTIVITY\;in\;mg\;QE\;Kg^{-1}\;min{\;}^{-1}}\;\begin{array}{l} =\;+2.99530\\ -9.20482\;\times \;10^{-4}\;\times \;\mathrm{MICROWAVE}\;\mathrm{POWER}\\ -0.19673\;\times \;\mathrm{WATER}\;\mathrm{TO}\;\mathrm{POMACE}\;\mathrm{RATIO}\\ +\;(3.78348\;\times \;10^{-4})\;\times \;\mathrm{EXTRACTION}\;\mathrm{TIME}\\ +\;(5.04735\;\times \;10^{-5})\;\times \;\mathrm{MICROWAVE}\;\mathrm{POWER}\;\times \;\mathrm{WATER}\;\mathrm{TO}\;\mathrm{POMACE}\;\mathrm{RATIO}\\ +\;(9.94101\;\times \;10^{-8})\;\times \;\mathrm{MICROWAVE}\;{\mathrm{POWER}}^{2}\\ +\;(3.07567\;\times \;10^{-3})\;\times \;\mathrm{WATER}\;\mathrm{TO}\;\mathrm{POMACE}\;{\mathrm{RATIO}}^{2}\\ +\;(2.31225\;\times \;10^{-5})\;\times \;\mathrm{EXTRACTION}\;{\mathrm{TIME}}^{2}\\ -\;(4.12374\;\times \;10^{-9})\;\times \;\mathrm{MICROWAVE}\;{\mathrm{POWER}}^{2}\;\times \;\mathrm{WATER}\;\mathrm{TO}\;\mathrm{POMACE}\;\mathrm{RATIO}\\ -\;(3.85059\;\times \;10^{-7})\;\times \;\mathrm{MICROWAVE}\;\mathrm{POWER}\;\times \;\mathrm{WATER}\;\mathrm{TO}\;\mathrm{POMACE}\;{\mathrm{RATIO}}^{2} \end{array}$$

Finally, the optimum extraction conditions and the corresponding maximum rate of extraction of raw OP total flavonoids were found to be the following:Microwave power: 2000.00 WWater to solid orange pomace ratio: 24.35Extraction time: 15.00 min

Optimum (maximum) rate of OP total flavonoids extraction (maximum OP flavonoids productivity at industrial scale) equaled 26.08 mg QE Kg^−1^ min^−1^.

### 2.5. Validation of the Mathematical Models Developed to Predict the Extracted Amount of OP Total Polyphenols and OP Total Flavonoids as Well as the Productivities of the OP VMAE Concerning Total Polyphenols and Flavonoids

For the validation of the precision of the four developed predictive mathematical models, the absolute average % error between the measured and predicted values were calculated and they are presented in the following Figure 9.

From the values of the % mean error of prediction for the four RSM models presented in the Figure 9, a high prediction accuracy is concluded for all models and especially in the case of the OP total polyphenols. In addition, a two-tailed paired t-test was performed between the actual and the predicted values of the antioxidant properties of the OP extracts as well as between the actual and the predicted rates of extraction of OP total polyphenols and total flavonoids and the results, which are presented in Table 7, suggest in all cases that there is no statistically significant difference between the actual and predicted values. This is yet another proof towards the robustness of the derived models.

## 3. Discussion

### 3.1. The Effect of Process Parameters on the Extracted Amounts of OP Total Polyphenols and Flavonoids by Industrial Scale VMAE Extraction

In the case of the extraction of OP total polyphenols, the factor A (Microwave power) has a significant and very clear increasing effect on the total amount of the extracted OP polyphenols, while the increase of factor B (Water to orange pomace ratio) has initially a positive effect on the extracted amount of total OP polyphenols up to a maximum value is reached and after that the extracted amount total OP polyphenols declines. Finally, as far as the extraction factor C (Extraction time) is concerned, this has a clear increasing effect on the amount of extracted total OP polyphenols. Similar trend concerning the effect of extraction parameters on the MAE extraction yield, reported by Özbek et al. [31] and Mendes et al. [32] in the case of MAE of polyphenols extraction from pistachio hull and *Lycium barbarum* and the explanation that was given by them is the following: (a) the increase in microwave power results in an increase in the extracted amount of total polyphenols which may be explained by the easier penetration of the solvent into the plant structure as a result of increasing power that facilitates cell rupture and thus enhances the extraction; (b) higher phenolic yield can be obtained at the longer extraction time and higher solvent-to-sample ratio and the extraction yield increases with the increase in solvent-to-sample ratio, but, when the ratio exceeded a certain limit, the amount of extracted total polyphenols began to decrease. This decrease that is observed after a limit of water-to-solid ratio is reached can be attributed to the excessive increase of the total volume exposed to the microwaves which reduces the wave effect per Kg of total mass. Moreover, the significant terms of the derived predictive model for the extracted amount of OP total polyphenols are: A, C, AB, AC, B^2^, C^2^, A^2^B and AB^2^ which, according to the p-data listed in Table 2, present p-values less than 0.05 and thus mark the significant terms and interactions of the extraction parameters that affect the model response value (amount of extracted OP total polyphenols).

In addition, in the case of extraction of OP total flavonoids factor A (microwave power) has a clear and substantial decreasing effect on the total amount of the extracted OP flavonoids, which can be attributed to an inherent sensitivity of OP flavonoids to high-intension MW treatment. Moreover, regarding factor B, initially, by increasing it, a substantial positive effect on the extracted amount of OP total flavonoids is observed up to a maximum value and afterwards the extracted amount of OP total flavonoids starts to decline by further increase of B factor. Finally, with regards to the extraction parameter C (extraction time) initially has a clear increasing effect on the amount of extracted total OP flavonoids until this reaches a maximum value. Then any further increase of extraction time causes a rapid reduction of the extracted amount of total OP flavonoids probably due to the sensitivity of the OP flavonoids to long exposure to the MW power and high temperatures. This finding is fully in line with the views expressed by [17] who have commented about the sensitivity and easy degradation of the flavonoids of orange juice solid waste in case they are exposed at either high extraction temperature or intensive extraction conditions (e.g., high microwave or ultrasound power) and they also confirmed by Das [33] who conducted MAE extraction of natural flavonoids from onion peels and found out the same trend as described above. Finally, the significant terms of the predictive model of the extracted amount of OP total flavonoids are: A, AB, AC, A^2^, B^2^, C^2^, A^2^B, AB^2^ and B^2^C which, according to the ANOVA statistics summarized in Table 3, present p-values less than 0.05 and thus mark the significant terms and interactions of the extraction parameters that affect and determine the model response value.

### 3.2. The Effect of Process Parameters on the Productivities of the Industrial Scale VMAE Extraction of Raw OP Total Polyphenols and Flavonoids

In the case of the rate of the VMAE extraction (productivity) of OP total polyphenols, by increasing the factor A (Microwave power), the rate of extraction of OP total polyphenols increases until it reaches a maximum value and after that it is getting reduced. Similarly, by increasing the factor B (Water/OP value), initially a positive effect on the rate of the extracted total OP polyphenols is observed up to a pick value and subsequently the rate of extraction of OP total polyphenols decreases. Conversely, as far as the extraction factor C (Extraction time) is concerned, as this increases it causes a remarkable very steep decrease of the rate of extraction of total OP polyphenols. This suggests that the maximized extraction rate of OP total polyphenols which provide the most economic operation (maximum total polyphenols productivity at industrial scale) is obtained at a very low extraction time (just 15 min). Moreover, the significant terms of the OP total polyphenols rate of extraction predictive model are A, C, AB, AC, BC, B^2,^ C^2^, A^2^B, A^2^C, AB^2^ and C^3^ which, according to the ANOVA statistics listed in Table 5, present p-values less than 0.05 and thus mark the significant model terms and interactions of the extraction parameters that affect and determine the model response value (total polyphenols productivity).

In addition, in the case of the rate of extraction (productivity) of OP total flavonoids the factor A (Microwave power) has a clear and substantial decreasing effect on the extraction rate of the OP flavonoids which can probably be attributed to an inherent sensitivity of OP flavonoids to high intension MW treatment. Moreover, concerning factor B (water to OP ratio), initially, by increasing B, a substantial positive effect is introduced to the rate of extraction of the OP total flavonoids and, then, as a certain maximum value is reached, afterwards the rate of extraction of OP total flavonoids starts to decline as factor B further increases. Finally, regarding parameter C (extraction time), by increasing this, a very steep decrease on the rate of extraction of total OP flavonoids is observed. This means that the maximum productivity of OP total flavonoids is obtained after a short interval (just 15 min) and low MW power. Finally, the significant terms of the extraction rate predictive model for OP total flavonoids are A, C, AB, A^2^, B^2^, C^2^, A^2^B and AB^2^, which, according to the data listed in Table 6, have p-values less than 0.05 and thus mark significant model terms and interactions of the extraction parameters that affect the model response value (total flavonoids productivity).

### 3.3. Comparison of the Optimized Values of Extracted OP Total Polyphenols and Flavonoids of the Present Research Work with the Results Corresponding to Previous Works

According to the results of the present study the maximum amount of the polyphenols extracted from raw OP by vacuum-microwave-assisted extraction (VMAE) is 13,559.802 mg per 2 Kg of raw orange pomace or, equivalently, 6780 mg GAE Kg^−1^ raw orange pomace. Furthermore, as the dry matter of raw OP was determined to be 18% *w*/*w*, the maximum extracted OP total polyphenols expressed on a dry basis is calculated to be 6780 × 100/18 = 37667 mg GAE Kg^−1^ of OP dry matter. Similarly the maximum amount of the extracted OP flavonoids expressed on a dry basis is 1909.27 mg QE/2 × 100/18 = 5304 mg QE Kg^−1^ of dry OP.

According to Putnik et al. [6], the maximum of the extracted total polyphenols from dry orange peel extracts falls in the range 9100–49,200 mg GAE Kg^−1^ of dry peel while the corresponding maximum of flavonoids falls in the range of 2000–30,000 mg QE Kg^−1^ of dry peel. Further, Dahmoune et al. [16] & Mhiri et al. [17] reported yields of total polyphenols extracted from orange dry peel in the range of 12,200–15,740 mg of GAE Kg^−1^ obtained by using as extraction solvent water/ethyl alcohol solutions. In addition, Londono-Londono et al. [15] cited that aqueous ultrasound-assisted extraction of dry orange pomace yielded a value of total polyphenols equal to 66,360 mg GAE Kg^−1^ of dry orange waste and, for total flavonoids, a value of 40,250 mg QE kg^−1^ but this was obtained by using lab-scale equipment and OP raw material in dry powder form.

Furthermore, Faber et al. [22] & Memduha et al. [23], by investigating the supercritical carbon dioxide extraction of orange pomace, found that the obtained yield of total polyphenols by this technology ranged from 18,000 to 27,800 mg GAE Kg^−1^ of dry OP depending on the experimental conditions. In addition, Anagnostopoulou et al. [34], reported extraction yield of total polyphenols from orange waste in the range 36.3–2540 mg GAE Kg^−1^ DM and Casquete et al. [20] 2840 mg GAE Kg^−1^ DM. Finally, Escobedo-Avellaneda et al. [35], by using as extraction material the external colored section of the orange fruit (usually called flavedo), achieved a total polyphenols yield of 5886–6799 mg GAE Kg^−1^ of DM. From the above mentioned literature references, it is concluded that, in general, the yield of the extracted total polyphenols from the orange juice industry solid waste lies in the wide range of 36.3–66,360 mg GAE Kg^−1^, whereas the yield of total flavonoids was reported to be in the range of 2000–40,250 mg QE Kg^−1^ of dry solid waste. It is also worth noting that, according to Barbosa et al. [36], these substantial differences among the reported values by various researchers concerning the total polyphenols and flavonoids concentrations as well as total antioxidant activity of the orange juice solid wastes extracts can be attributed to the variations of the raw material (different cultivars and solid tissue composition) and to the use of different solvents and extraction modes.

Comparing the literature values with the findings of the present work, it is concluded that our optimum value for OP total polyphenols yield obtained by VMAE extraction is slightly above the middle of the range reported in the literature (37,667 mg GAE Kg^−1^ of dry OP compared to the highest value of 66,360 mg GAE Kg^−1^ reported in the literature) whereas the optimum value of the yield of OP total flavonoids of 5304 mg QE Kg^−1^ of dry OP obtained by VMAE extraction in the present work is within the range 2000–42,250 mg QE Kg^−1^ of dry OP reported in the literature. This can be attributed first to the quality of the raw OP used in the present work, which is substantially different to the dry and finely comminuted orange peel used in almost all cases by other researchers and also to the mechanical removal of the orange essential oil before the mashing of the fruits to obtain the juice by our supplier of the OP. This operational mode removes the major part of the natural antioxidants that originally are contained in the orange peel in the form of essential oil and therefore reduces the concentration of the phenolics in the orange pomace. In addition, the orange cultivar and the extractor dynamics can be considered as parameters that could cause this difference. Finally, in the present research, for the first time, according to our knowledge, optimization of the aqueous VMAE for raw orange pomace at industrial scale was carried out. For this reason, the results obtained in the context of the present work cannot be directly comparable to those of previous research works because all of them were conducted at lab scale. However, our results are particularly interesting and readily usable by the industry as no scale up is required for them.

### 3.4. Comparison of the Model Derived in the Present Work with Models Suggested in the Literature

The vacuum-microwave-assisted extraction is a typical paradigm of a solid–liquid extraction and many researchers have developed models to be used for the prediction of the concentration or alternatively of the amount of the extracted bioactive compounds. As cited by Akhtar [37], whenever working with microwave-assisted extraction, the efficiency of the process can be enhanced by considering the solvent type, solvent to plant material ratio, extracting power of microwaves, resultant temperatures, time of extraction as well as nature of plant matrix and targeted molecule. We can have significant results only by considering the above mentioned factors in microwave-assisted extraction [38].

In the literature, there are two types of mathematical models dedicated to microwave extraction: *theoretical models* based on chemical engineering principles about solid–liquid diffusion [39,40,41,42] and *empirical statistical models* based mostly on Response Surface Methodology (RSM) [17,31,43,44,45] but also on Adaptive Neuro-Fuzzy Inference System (ANFIS) statistical methodology [46,47].

From the above-mentioned models, the theoretical ones are not very useful in the case we need to obtain the optimum conditions for industrial-scale optimization of the VMAE extraction of bioactive phytochemicals. This is because they involve only the dependence of the extracted amount of the targeted substance vs. time and not the combined effect of the three significant extraction parameters (Microwave power, water to solid ratio and extraction time). On the contrary, the empirical statistical models connect the extraction yield to all extraction parameters and they can give the overall optimum of the response. Furthermore, the empirical models incorporate the effect of the disintegration of a part of the total polyphenols and flavonoids due to shear or thermal stress during the process which is not taken into account in all of the above-mentioned theoretical models.

For the above reason, in our case, a novel empirical model was selected based on the RSM methodology but with significant differences compared to the typical Box and Behnken experimental design used by previous researchers. In particular, with the target to increase the accuracy of the optimization, a denser experimental plan designated as “historical data design” was adopted with a significantly larger number of experimental points compared to the well-known Box & Behnken design used by other researchers. On top of this, by using the Cox & Box plot provided by the statistical software (Design Expert), a proper transformation of the model response was selected in order to improve the model accuracy and, in addition, a cubic instead of quadratic polynomial model was used to fit the experimental data. The increase of the degree of the polynomial model from 2 to 3 is suggested by the Design Expert software in cases where there is a strong nonlinearity in the dependence of the response (amount of extracted phytochemical) on the extraction parameters. This high nonlinearity can be easily concluded if we observe the form of theoretical model equations given in the literature above.

However, the novel approach of predictive modeling and optimization given in this paper can be tested further with more applications in order to be validated. Then it can be used to increase the effectiveness of the derived empirical statistical models dedicated to microwave extraction, by using a larger number of experimental points which can eliminate the negative effect of a single erroneous measurement and by proper transformation of the response. 

Finally, despite the fact the experimental effort is heavier by this approach because of the higher number of experiments and the industrial size, the much better precision and the avoidance of scale-up can pay back for the extra effort.

### 3.5. Summary of the Points of Novelty of the Present Research Work

Concerning the novelty of the present research, this is based on the following points: (a) predictive modeling and optimization of OP extraction by aqueous “green” vacuum-microwave-assisted extraction for the first time at industrial scale and involving both maximum recovery as well as economic criteria; (b) use of raw orange pomace against dried used in previous research works and therefore avoidance of the costly and quality-deteriorating pomace drying and milling pretreatments; (c) use of statistical criteria for the correct selection of the optimum degree of the RSM polynomials and the optimum transformation of the dependent variables; and (d) use a comprehensive and multipoint “historical data” experimental design involving 80 experimental points in total in order to improve the accuracy of the derived predictive models and the precision of the determination of the optimum values. 

## 4. Materials and Methods

### 4.1. Orange Pomace

The orange pomace was kindly supplied by the Greek orange juice producer, ALBERTA S.A. which is established in Argos Peloponnese-Greece. The orange variety from which the obtained pomace was coming from was the well-known orange fruit variety “Navel”. The obtained pomace was passed through a commercial meat mincer (model CANDY COMET supplied by D. Tomporis Co, 92 Cyprus str, Larisa, Greece) with a 3 mm hole diameter screen in order to become comminuted and then it was kept in a properly sealed plastic vacuum bag (2 Kg per bag) at −25 °C until used for extraction. Drying was not applied to the orange pomace in order to avoid oxidative degradation of the bioactive compounds.

### 4.2. Description of the Microwave Extractor and of the Extraction Methodology

The extraction of the orange pomace samples was conducted by using the industrial-scale vacuum-microwave extractor model MAC-75 (Milestone Inc., Sorisole (BG)–Italy) which is established in the premises of PELLAS NATURE Co (Edessa, Greece) and is illustrated in Figure 10.

The extraction trials of the orange pomace samples were conducted following the procedure described below. The frozen orange samples were first thawed at ambient temperature and 2 Kg of each sample were then collected and used as the extraction sample. The 2 Kg orange pomace sample was first put in a plastic basket which was then adjusted to the extraction cavity of MAC-75 vacuum-microwave extractor. Consequently, the machine door was closed and filled with the appropriate quantity of distilled water. The quantity of the water used in each trial was according to the water/solid ratio suggested by the experimental plan (shown in Table 1). In addition, the desired values of microwave power and extraction time were set via the electronic panel of the extractor, according to the experimental plan, and the industrial scale extractor was set in automatic operation. Cooling was not used during the extraction period and the temperature set point was set to maximum 80 °C (adopting a rising temperature extraction mode of operation). During each extraction trial, the samples of the extracts were collected at regular time intervals (15 min, 30 min, 45 min, 60 min, 75 min, 90 min and 120 min), filtered through plain filter paper and the filtrates were collected in plastic bottles and coded accordingly in order to easily and safely distinguish different samples. The collected samples were kept frozen at −25 °C in the freezing facility of the Laboratory of Food and Biosystems Engineering (University of Thessaly) for a short period until the selected bioactivity parameters were analyzed.

### 4.3. Total Polyphenols Determination Method

For the determination of the total polyphenols as GAE (gallic acid equivalents) of the obtained orange extracts, a slightly modified version of the method of Singleton et al. [48] and Waterhouse [49] was used. According to this method, initially a gallic acid solution was prepared by dissolving 0.5 g gallic acid in 10 mL of pure ethanol and the solution was then transferred in a 100 mL volumetric flask and the rest of the volume was filled by distilled water (preparation of a gallic acid stock solution of 5000 ppm). In addition, in a 1 L glass beaker, 200 g of anhydrous sodium carbonate were dissolved in 800 mL distilled water and the solution was boiled until the salt was fully dissolved. The solution was then cooled and kept at 24 h in dark, which resulted in the formation of crystals of anhydrous sodium carbonate, which were removed by filtration the next day. The clear filtrate was finally dissolved in a total volume of 1 L by adding the remaining distilled water in a 1L volumetric flask. Consequently, a set of standards of gallic acid was prepared by diluting 0 mL, 1 mL, 2 mL, 3 mL, 5 mL, 10 mL, and 20 mL of the gallic acid stock solution in six volumetric flasks of 100 mL each and filled with distilled water up to 100mL volume in order to prepare standard solutions of 0, 50, 100, 150, 250, 500 and 1000 ppm gallic acid. From each standard solution a quantity of 20 μL was mixed with 1.58 μL distilled water and 100 μL Folin–Ciocalteu reagent in a glass tube, and, within 8 min, a quantity of 100 μL sodium carbonate solution was added and the tubes were incubated for 2 h at 20 °C, after which their absorbance was measured by a UV-Vis photometer (model EVOLUTION TM 201 supplied by Thermo-Scientific Co, Shanghai, China) against the blind solution (0 ppm gallic acid concentration). The standard curve depicting gallic acid concentration vs. absorbance was constructed using the Microsoft Excel software and its R^2^ value was 0.9982. Calculation of the total polyphenols of extracts of OP was carried out following the same procedure and using the following equation of the standard curve:Total polyphenol concentration of extract in ppm of GAE = absorbance of sample at 765 nm/0.001(5)

Each measurement concerning total polyphenols was carried out in triplicate and the result was the average of the three obtained values.

### 4.4. Total Flavonoids Determination Method

The total flavonoids content expressed as mg of quercetin equivalents (QE)/L of the obtained orange pomace extracts was determined by using the colorimetric method of AlCl_3_, as described by [50]. The method is based on the principle that AlCl_3_ reacts with the hydroxyls of the flavonoids and produces a colored complex which has maximum absorbance at 420 nm. The total flavonoids content was expressed as quercetin equivalents (QE) per L of extract. The determination method for the total flavonoids was carried out as follows: 1.0 mL of the orange pomace extract or standard solution (used for the construction of the calibration curve) was added in a glass test tube to which 3 mL methanol, 200 μL of aqueous solution of 10% *w*/*v* AlCl_3_, 200 μL 1Μ potassium acetate solution and 5.6 mL distilled water were added. The tube was then agitated by vortex and incubated for 30 min at ambient temperature for the completion of the chemical reaction. The absorbance of each sample was measured at 420 nm against a blind solution which contained all the reagents except for the orange pomace extract which was replaced by distilled water. 

For the construction of the calibration curve, a quercetin stock solution of 1000 ppm was prepared as well as a series of standard solutions of 50, 100, 200, 500 and 1000 ppm by serial dilutions of the stock. The absorbance of standard solutions was measured and plotted against their concentration and the linear equation obtained by Excel was used for the determination of the concentration of the total flavonoids of the OP extracts. The R^2^ value of the obtained linear correlation was 0.9834.

Calculation of the total flavonoids of OP extracts was carried out following the same procedure and using the following equation of the standard curve:Total flavonoids concentration mg QE/L of extract = absorbance of sample at 420 nm/0.0055(6)

Each measurement concerning total flavonoids was carried out in triplicate and the result was the average of the three obtained values.

### 4.5. Chemicals Used for Antioxidant Tests

All the chemicals used for the above mentioned antioxidant tests were selected from the standard catalog of the Sigma Aldrich company and they were supplied by the Greek representative Life Sciences Chemilab, 33 Amarantou str, 56431 Thessaloniki, Greece.

### 4.6. Modeling and Optimization Methodology

The methodology used for modeling and optimization had the following aims:Modeling and optimization of total orange pomace polyphenol extraction;Modeling and optimization of total orange pomace flavonoids extraction;Simultaneous optimization of total orange pomace polyphenols, total flavonoids;Modeling and optimization (maximization) of the rate of extraction of orange pomace polyphenols to achieve the maximum productivity (economic optimum);Modeling and optimization (maximization) of the rate of extraction of orange pomace flavonoids to achieve the maximum productivity (economic optimum).

A multipoint historical data experimental design employing 80 experimental points evenly spread in the design space was used along with response surface methodology (RSM) to derive the relevant polynomial mathematical models for the prediction of total polyphenols and total flavonoids of the orange pomace extracts and obtain the corresponding optimum values. Three factors were used as optimization factors: (a) the microwave power in the range of 2000 to 6000 W; (b) the ratio of extraction water to orange pomace in the range from 10 to 30; and (c) the extraction time in the range of 15 min to 120 min. There were five optimization responses: (a) total polyphenol content; (b) total flavonoids content; (c) total polyphenols and total flavonoids content simultaneously; (d) the rate of the extraction of the total orange pomace polyphenols; and (e) the rate of extraction of the total orange pomace flavonoids. The Design Expert 12.0.0 statistical software was used to preform predictive modeling and optimization and derive the mathematical models. The selection of the appropriate order for the polynomial models to fit the experimental data was based to the statistical evaluation tools of the Design Expert software and concerning the optimum response transformation in order to obtain satisfactory fitting this was selected in all cases according to the suggestion of the Cox & Box Diagram. The reliability of the obtained models was validated by statistical analysis (ANOVA) and in all cases the statistical significance of the derived models as well as the desirable nonsignificance of lack of fit were confirmed. The validity of the derived models was also tested by conduction of measurements at experimental points within the design space different than the ones used to develop the models and, consequently, by comparison of the obtained predicted and measured values and determination of the magnitude of the mean prediction error.

### 4.7. Determination Method of the Extraction Rates of OP Total Polyphenols and Flavonoids

The most interesting target of optimization for the industry, on economic grounds, is the maximization of the rate of extraction of polyphenols or flavonoids from raw orange pomace. 

The rate of the extraction of either total orange pomace polyphenols or total orange pomace flavonoids is determined using the following equations:(7)$$EXTRACTION\;RATE\;OF\;OP\;TOTAL\;POLYPHENOLS\;\left(RTPE\right)\left(mg\;GAE\;Kg^{-1}OP\;min^{-1}\right)\;\;=\frac{Amount\;of\;extracted\;OP\;total\;polyphenols\;in\;mg\;GAE}{\left(OP\;mass\;in\;Kg\right)\;x\;\left(t+tdelay\;in\;min\right)}$$
where in our case:-The amount of the extracted total orange pomace polyphenols as well as the corresponding extraction time (t) are given in Table 1;-The mass of the extracted raw OP was in all cases equal to 2 Kg;-Furthermore, the delay time (t_delay_) between successive extraction cycles was 15 min.

In a similar manner:(8)$$EXTRACTION\;RATE\;OF\;OP\;TOTAL\;FLAVONOIDS\;\left(RTFE\right)\;\left(mg\;QE\;Kg^{-1}OP\;min^{-1}\right)\;\;=\frac{Amount\;of\;extracted\;OP\;total\;flavonoids\;in\;mg\;QE}{\left(OP\;mass\;in\;Kg\right)\;x\;\left(t+\;tdelay\;in\;min\right)}$$
where:-The amount of the extracted OP total flavonoids is given in Table 1 as well as the corresponding extraction times (t);-The mass of extracted raw OP was in all cases equal to 2 Kg;-As above, the delay time (t_delay_) between successive extraction cycles was 15 min.

By using the above-described equations, the rates of extraction of OP total polyphenols and total flavonoids were calculated using the data listed in Table 1 with the purpose to be maximized by RSM methodology to obtain the specific conditions for economically optimum industrial scale operation. 

## 5. Conclusions

In the present work, for the first time, the aqueous eco green vacuum-microwave-assisted extraction of phenolics from bioactive raw orange pomace was investigated and optimized at real industrial scale. The essential difference of the present study in comparison with previous studies is that fresh humid raw pomace was used as extraction material instead of the dried orange pomace or dry peel in powder form which were used in the previous studies. This can improve the economics of potential industrial production of orange extracts because it omits the costly drying and milling steps involved in the case that dry orange pomace or peel is used as raw material for the extraction. According to the results obtained in the present study, natural antioxidant extracts can be produced at industrial scale by eco green vacuum-microwave extraction from fresh orange pomace at conditions optimized by Response Surface Methodology based on a novel multipoint historical data experimental design and using cubic polynomial models with appropriately transformed responses to improve the accuracy of the prediction. The optimum extraction yield expressed on dry orange pomace basis were found to be 37,667 mg of GAE Kg^−1^ and 5304 mg QE Kg^−1^ on a dry basis for OP total polyphenols and total flavonoids respectively.

The corresponding optimum values of the extraction parameters to achieve the above-mentioned maximum polyphenol and flavonoids extraction yields were found to be: (a) Microwave power = 5999.997 W, Water/OP ratio = 26.09 and Extraction time = 120.00 min for total polyphenols, and (b) Microwave power = 2000.00 W, Water/OP ratio = 24.12 and Extraction time = 53.45 min for total flavonoids. It is also worth noting that a set of extraction parameters corresponding to Microwave power = 2000.00 W, Water/OP ratio = 24.95 and Extraction time = 63.96 min could be an effective compromise in order to simultaneously optimize the total polyphenols and the total flavonoids of the orange pomace extract. Yet another important observation is that, in order to achieve the optimum (maximum) yield of polyphenols, operation at high microwave power and long time intervals, is required, while the optimum flavonoids yield is obtained at low microwave power and moderate time probably due to the sensitivity of their chemical structure.

Moreover, as industrial reality demands high productivity in order to achieve cost effective production, in the course of the present work the productivities of the VMAE extraction of raw OP total polyphenols and raw OP total flavonoids were optimized and the optimum conditions for total OP polyphenols and total OP flavonoids respectively were found to be (a) Microwave power = 4484.777 W, Water/OP ratio = 19.619 and Extraction time = 15 min for total polyphenols, and (b) Microwave power = 2000 W, Water/OP ratio = 24.35 and Extraction time = 15 min for total flavonoids. The optimization of the productivities can be used as explained for financial reasons while the maximization of the amounts of the extracted OP total polyphenols and flavonoids can alternatively be useful in case the emphasis is on the minimum content of phenolics in the postextraction waste. Something like that would be very important in the case that this waste is to undergo further processing (biorefinery principle) by fermentation in order to avoid undesirable retard of the used starter cultures by the remaining phenolics postextraction.

## Figures and Tables

**Figure 1 molecules-26-00246-f001:**
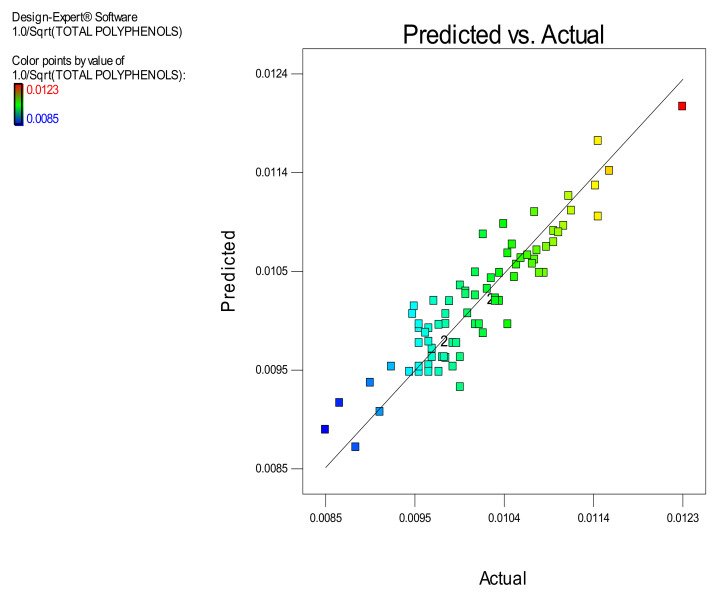
Correlation of Predicted vs. Actual values of the amount of the extracted orange pomace (OP) total polyphenols (mg GAE).

**Figure 2 molecules-26-00246-f002:**
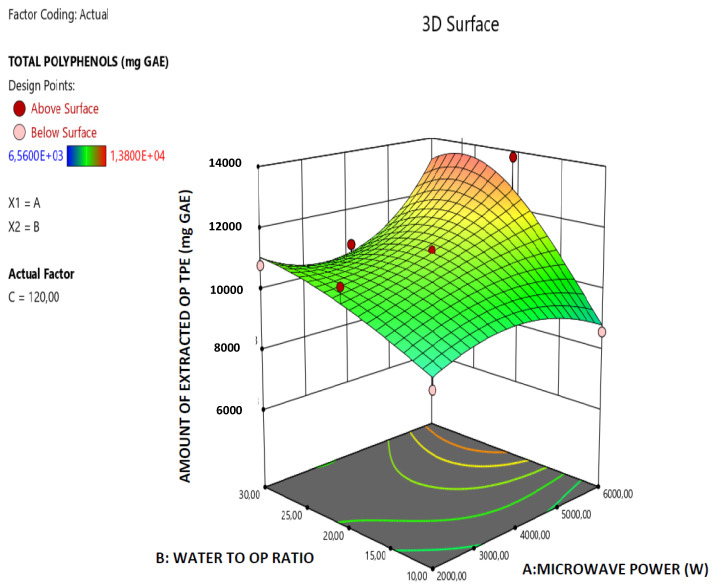
Response surface plot of the amount of extracted total OP polyphenols (TPE in mg GAE) A × B interaction (A: Microwave Power (W); B: water/OP ratio).

**Figure 3 molecules-26-00246-f003:**
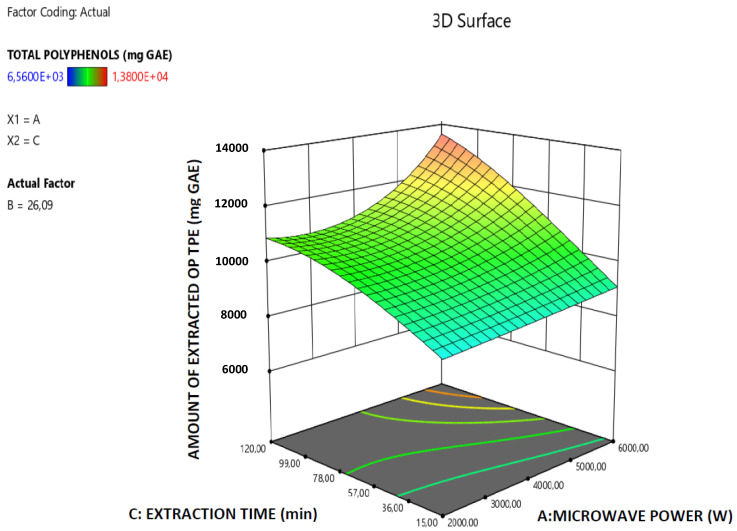
Response surface plot of the amount of extracted total OP polyphenols (TPE in mg GAE) A × C interaction (A: Microwave Power (W); C: extraction time (min)).

**Figure 4 molecules-26-00246-f004:**
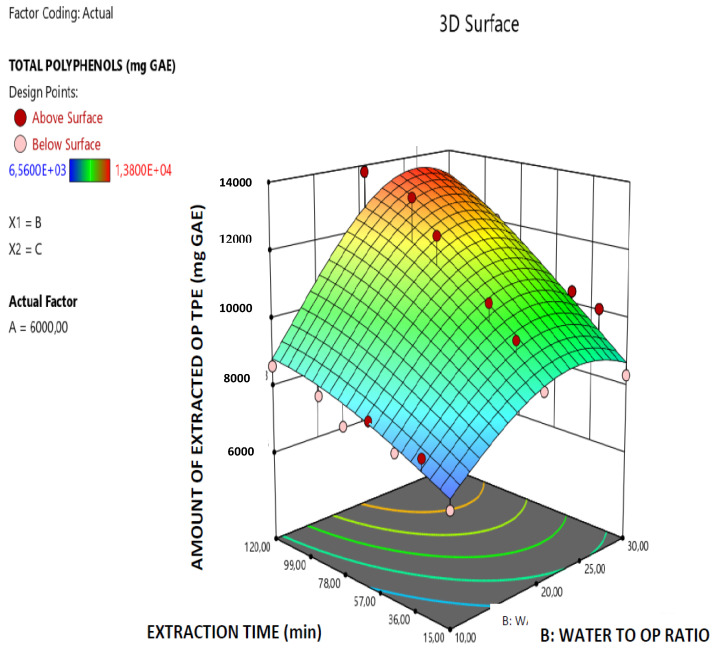
Response surface plot of the amount of extracted total OP polyphenols (TPE in mg GAE) B × C interaction (B: Water/solid ratio; C: Extraction time (min)).

**Figure 5 molecules-26-00246-f005:**
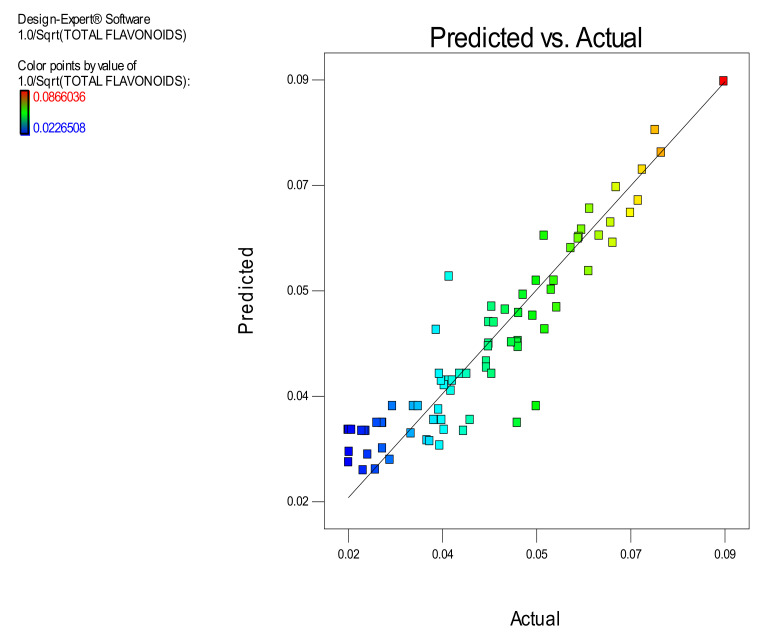
Correlation of Predicted vs. Actual values of the amount of extracted orange pomace total flavonoids.

**Figure 6 molecules-26-00246-f006:**
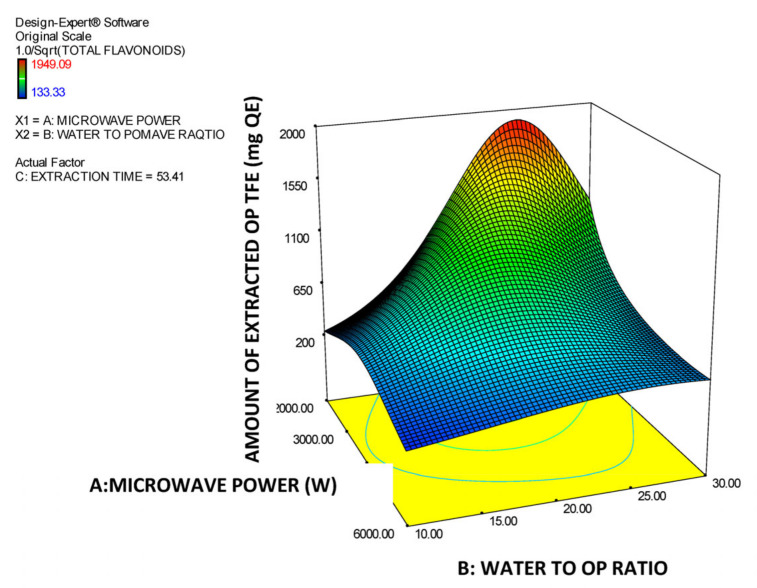
Response surface plot of the amount of extracted total OP flavonoids (TFE in mg QE) for the A × B interaction (A: Microwave power (W); B: Water to OP ratio).

**Figure 7 molecules-26-00246-f007:**
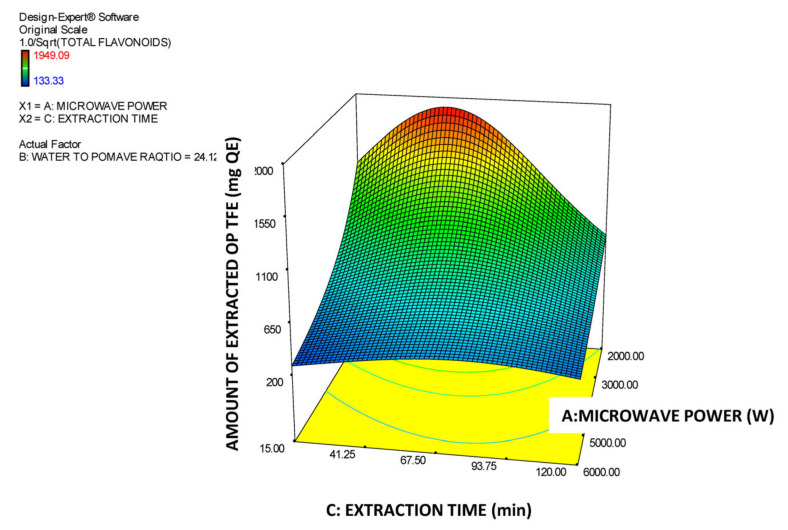
Response surface plot of the amount of extracted total OP flavonoids (TFE in mg QE) for the A × C interaction (A: Microwave power (W); C: Extraction time (min)).

**Figure 8 molecules-26-00246-f008:**
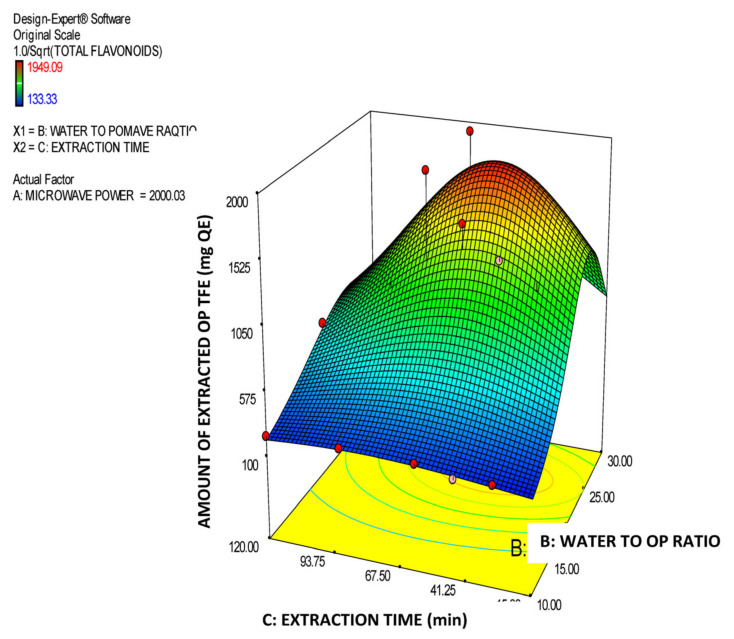
Response surface plot of the amount of extracted total OP flavonoids (TFE in mg QE) for the B × C interaction (B: Water to orange pomace ratio; C: Extraction time (min)).

**Figure 9 molecules-26-00246-f009:**
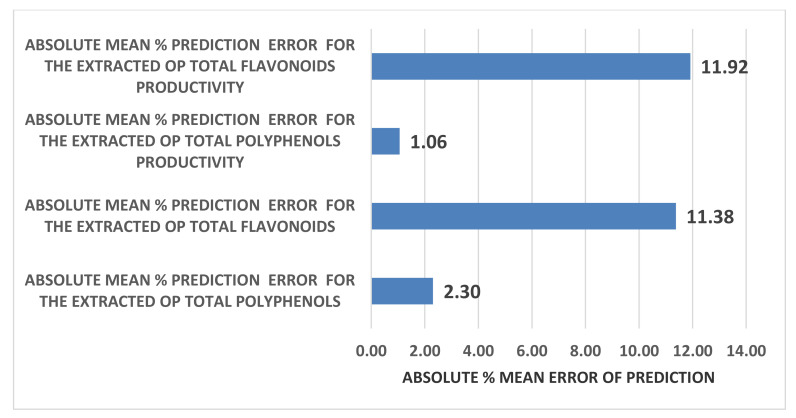
Absolute % mean error of prediction of the derived four Response Surface Methodology (RSM) models.

**Figure 10 molecules-26-00246-f010:**
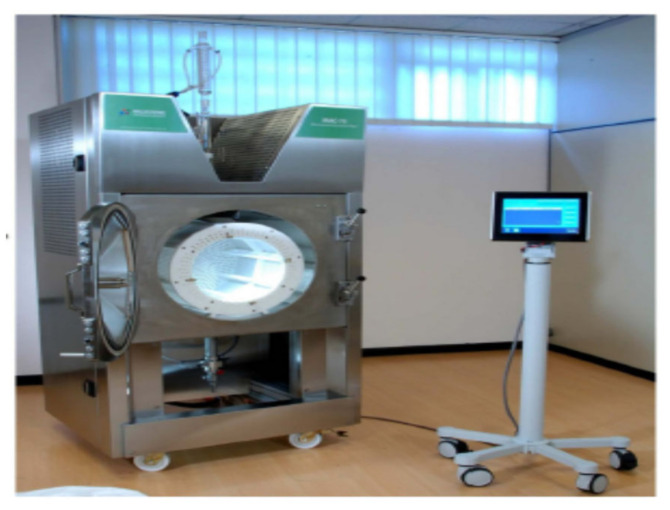
The setup of the industrial scale microwave extractor MAC 75/Μilestone Technologies.

**Table 4 molecules-26-00246-t004:** Optimum (maximum) values of antioxidant parameters of OP extracts vs. VMAE parameters.

	Optimized Extraction Conditions			Optimized Antioxidant Parameters	
Optimization Target	Microwave Power (W)	Water/OP Ratio	Extraction Time (min)	Maximum of Extracted OP Total Polyphenols (mg GAE per 2 Kg Raw OP)	Maximum of Extracted OP Total Flavonoids (mg QE per 2 Kg Raw OP)
Total polyphenols	5999.997	26.09	120.00	13,559.802	
Total flavonoids	2000.00	24.12	53.45		1909.27
Total polyphenols + Total flavonoids	2000.00	24.95	63.96	9977.48	1847.42

**Table 7 molecules-26-00246-t007:** Results of the statistical t-test analysis applied between predicted and actual values of the extracted total polyphenols and total flavonoids of orange pomace extracts as well as between the predicted and actual productivities of them.

a/a		t Value	*p* Value	Statistical Significance (2-Tailed) Significance Level = 0.05
1	Pair 1 PM-PP	−0.17411	0.862004	*p* value = 0.862004 > 0.05 therefore no significant difference between predicted and measured amounts of OP total polyphenols
2	Pair 2 FM-FP	−0.02704	0.978465	*p* value = 0.978465 > 0.05 therefore no significant difference between predicted and measured amounts of OP total flavonoids.
3	Pair 3 PRM-PRP	0.01359	0.989174	*p* value = 0.989174 > 0.05 therefore no significant difference between predicted and measured extraction rate values of OP total polyphenols
4	Pair 4 FRM-FRP	−0.01204	0.990407	*p* value = 0.990407 > 0.05 therefore no significant difference between predicted and measured extraction rate values of OP total flavonoids
	**(1) PM**—measured values of extracted total OP polyphenols **(2) PP**—predicted values of extracted total OP polyphenols **(3) FM**—measured values of extracted total OP flavonoids **(4) FP**—predicted values of extracted total OP flavonoids **(5)****PRM**—measured values of the rate of extracted total OP polyphenols **(6)****PRP**—predicted values of the rate of extracted total OP polyphenols **(7)****FRM**—measured values of the rate of extracted total OP flavonoids **(8)****FRM**—predicted values of the rate of extracted total OP flavonoids			

## Data Availability

The data presented in this study are available on request from the corresponding author.

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
