# Peer review of "Optimization of Vacuum-Microwave-Assisted Extraction of Natural Polyphenols and Flavonoids from Raw Solid Waste of the Orange Juice Producing Industry at Industrial Scale"

_molecules, 2021, doi:10.3390/molecules26010246_

Round 1

Reviewer 1 Report

COMMENTI

The Molecules-1040482 manuscript titled "Optimizing the Vacuum Microwave Assisted Extraction of Natural Polyphenols and Flavonoids from Raw Solid Waste of the Industrial Scale Orange Juice Industry" by Konstantinos Petrotos al., Is a research article and its purpose is to investigate and optimize on an industrial scale, and with economic criteria, the microwave extraction of raw orange pomace and obtain the optimal conditions for obtaining the maximum yield and productivity of polyphenols and / and flavonoids.

The article investigates an interesting topic. However, some points need to be overcome.

The authors use deep green eutectic solvents (DES), natural deep eutectic solvents (NDES) for their extraction protocol:
Explain if the ecotoxicological properties of DES used with the applied extraction technique are known.
Can used DES be recycled? If so, does DES regeneration cause a decrease in efficiency?
Does the polyphenol extraction protocol used provide for the formation of by-products?

Can the aqueous eco-green vacuum assisted microwave extraction of phenols from bioactive raw orange pomace reduce the antioxidant power of polyphenols? Authors should respond, please.
Specific comments:

Figure 1: Add comments in the Results section.

Author Response

Please find attached our response to Reviewer1

With respect

Prof. Konstantinos Petrotos

Reviewer 2 Report

This paper makes use of RSO to optimize and industrial-scale process for extraction of polyphenols and flavonoids. The methodology is sound and appropriate. However, the following aspects need to be clarified and elaborated further before accepting the manuscripts for publication:

  1. Given that this paper it is to be published in a journal about "molecules", the authors should mention the nature of the polyphenols and flavonoids extracted, or provide some examples of the molecules contained in the extracts.
  2. The methodology used is empirical and can be specific to the system under study and is less likely to be applicable to other cases. But it can be useful to get some insights about the process. The authors should elaborate further on what parameters influence the extraction the most and the possible physical/chemical reasons behind it, and suggest potential future research on those aspects.
  3. Your technology can definitely help orange juice processors to improve the treatability of their wastes and mitigate some wastewater treatment issues. Furthermore, can help to improve the energy-water nexus of citrus processors as suggested in previous relevant work. Please refer to "Martinez-Hernandez, E., Molina, M. M., Flores, L. A. M., Ruiz, M. E. P., EguiaLis, J. A. Z., Molina, A. R., ... & Amezcua-Allieri, M. A. (2019). Energy-water nexus strategies for the energetic valorization of orange peels based on techno-economic and environmental impact assessment. Food and Bioproducts Processing117, 380-387."
  4. Is it possible to compare or mention the advantages of your modelling approach to other empirical or first-principle modelling approaches?
  5. Is it possible in your approach to perform feature/variable selection so that your model can be reduced to only relevant variables while keeping reasonable prediction capability?

Author Response

Please find attached our response to review comments.

With respect

Prof. Konstantinos Petrotos

Reviewer 3 Report

Title of submitted manuscript is too long and a suggestion as to how it might sound is: “Vacuum microwave assisted extraction of orange pomace natural polyphenols and flavonoids at industrial scale – optimization”. It does not commonly contain full stop at the end.

Abstract is well written with recommendation to separate factors used for the optimization process and mentioned statistical analysis.

Introduction is too long and not very focused. Here is necessary explain the background of research. It could cover two directions: waste utilization as a source of biologically active compounds and process sustainability in the food and pharmaceutical industries with highlight of eco-friendly aqueous vacuum microwave system. Now, as it is, it looks more like a discussion. Based on reformulated and concise introduction, it would be desirable to formulate more detail specify hypothesis which will be answered through in discussion and finally in conclusion. Lines 34 and 54: preposition in replace with to; Lines 51-52: match lowercase and uppercase letters; Line 57: did you mean chlorogenic acid. Line 129: extra unnecessary bracket. Through whole text check double spaces and add spaces on places where it is required (write uniformly).

Results and Discussion, among other parts, must contain disadvantages of done study which is observed by the researchers themselves. Regarding Table 1, it contains two asterisks without explanation. In addition total polyphenol and flavonoid content should be expressed in “complete” units like mg GAE/ kg fresh weight or mg QE/ kg dry matter… You have to precise how you want to show your results (mean value ± SD or mean value ± SEM) whereby it is necessary to pay attention on decimal numbers and to choose right number of decimals for right numbers and units. Each stated statistical value should be followed by a p value (including R2). Figure 1 and 5 should be shown with more details; at least axes have to be named and have to have units (they are not mentioned in the text). Figures 2, 3, 4, 6, 7, 8 represent polyphenol/flavonoid content expressed as mg GAE/QE per what? Figure 6 has no good quality (replace it). Line 273: you describe Table 3 but state Table 2. Line 426: you mean Figure 9 but state Figure 10. Line 505: number 10607,1 should be replaced with 10607.1. Most of results are shown on 2 kg of waste, why? Last sentence in the discussion part is long and confuse, thus it is advisable to separate it into two shorter and concise observations.

Experimental part is written well with indication that method for total polyphenols determination is lack of samples analysis. You did spectrophotometric methods, why you do not mentioned calibrations curves equations? A section on statistical analysis has to be described. Line 565: pomegranate or orange samples? Line 592: two full stops; Equation 8: unit for flavonoids is not mg GAE;

Conclusion is usually written in one paragraph and includes the most important contributions of the paper without repeating the conclusions mentioned above. So, it should be reduced and more actual. Line 734: reported by various research groups is redundant part of the sentence. Lines 737-740: reformulate in order to be clear.

Author Response

Please find attached our reply to the review comments.

With respect

Prof. Konstantinos Petrotos

Round 2

Reviewer 2 Report

The authors have done a good job in the revision.

Reviewer 3 Report

I have taken a look at point-to-point answers to my suggestions and can conclude that the authors approached the refinement of their manuscript carefully. The only thing left for them is to correct the last two columns in Table 1 according to the adequate number of decimals, as well as to follow the rule that the mean value and SD usually have the same number of decimals.